# Fractionation of *Carlina acaulis* L. Root Methanolic Extract as a Promising Path towards New Formulations against *Bacillus cereus* and Methicillin-Resistant *Staphylococcus aureus*

**DOI:** 10.3390/molecules29091939

**Published:** 2024-04-24

**Authors:** Sylwia Wnorowska, Agnieszka Grzegorczyk, Jacek Kurzepa, Filippo Maggi, Maciej Strzemski

**Affiliations:** 1Department of Medical Chemistry, Medical University of Lublin, 20-093 Lublin, Poland; jacek.kurzepa@umlub.pl; 2Department of Pharmaceutical Microbiology, Medical University of Lublin, 20-093 Lublin, Poland; agnieszka.grzegorczyk@umlub.pl; 3Chemistry Interdisciplinary Project (ChIP), School of Pharmacy, University of Camerino, Via Madonna Delle Carceri, 62032 Camerino, Italy; filippo.maggi@unicam.it; 4Department of Analytical Chemistry, Medical University of Lublin, 20-093 Lublin, Poland

**Keywords:** natural products, plant-derived material, antimicrobial activity, *Carlina acaulis* L.

## Abstract

The root of *Carlina acaulis* L. has been widely used in traditional medicine for its antimicrobial properties. In this study, the fractionation of methanol extract from the root was conducted. Four fractions (A, B, C, and D) were obtained and tested against a range of bacteria and fungi. The results showed promising antibacterial activity, especially against *Bacillus cereus*, where the minimal inhibitory concentration (MIC) was determined to be equal to 0.08 mg/mL and 0.16 mg/mL for heptane (fraction B) and ethyl acetate (fraction C), respectively. In the case of the methicillin-resistant *Staphylococcus aureus* (MRSA) ATCC 43300 strain, the same fractions yielded higher MIC values (2.5 and 5.0 mg/mL, respectively). This was accompanied by a lack of apparent cytotoxicity to normal human BJ foreskin fibroblasts, enterocytes derived from CaCo2 cells, and zebrafish embryos. Further analyses revealed the presence of bioactive chlorogenic acids in the fractionated extract, especially in the ethyl acetate fraction (C). These findings support the traditional use of the root from *C. acaulis* and pave the way for the development of new formulations for treating bacterial infections. This was further evaluated in a proof-of-concept experiment where fraction C was used in the ointment formulation, which maintained high antimicrobial activity against MRSA and displayed low toxicity towards cultured fibroblasts.

## 1. Introduction

Microbial resistance to antibiotics is becoming one of the most serious challenges to global healthcare systems, and the scale of this problem may be larger than that of HIV and malaria. In 2019 alone, the estimated global number of deaths associated with antimicrobial-resistant (AMR) microorganisms, mainly *Escherichia coli*, *Staphylococcus aureus*, and *Pseudomonas aeruginosa*, was 4.95 million [1]. Methicillin-resistant strains of *S. aureus* (MRSA) are the cause of increasingly frequent nosocomial and non-hospital infections of, e.g., the respiratory tract, bones, joints, and skin. The treatment of these infections is difficult and generates significant costs. It typically involves the use of teicoplanin and vancomycin (internally), while an ointment containing mupirocin is used externally at a concentration of 2%. However, it has been shown that MRSA resistance to mupirocin builds up very quickly due to its frequent use. In such a situation, the introduction of a new formulation containing substances with activity against MRSA that have not been used in the eradication of these microorganisms so far is highly desirable [2]. Yeasts of the genus *Candida*, especially *C. albicans*, are commensal fungi found in humans, especially in the oral cavity, gastrointestinal tract, and vagina. However, in immunocompromised individuals, they can be opportunistic pathogens causing superficial and systemic candidiasis. It is estimated that one in four cases of candidiasis is mixed bacterial–fungal in nature, with *S. epidermidis* and *S. aureus* being the most common pathogens co-occurring with *C. albicans* [3]. It seems important, therefore, to look for new substances that can be used to treat such mixed infections. *Bacillus cereus* and *Salmonella* infections are common causes of gastrointestinal and systemic infections [4,5]. They are treated with antibiotics, including vancomycin [6], and chemotherapeutics, such as ciprofloxacin [7]. However, such treatment is not always desirable, as in the case of gastrointestinal salmonellosis, where the use of antibiotic therapy sometimes causes *Salmonella* carriage [4] and the frequent use of antibiotics leads to the development of antibiotic resistance [8].

The effectiveness of antibiotics, which have saved millions of lives in the past, is drastically declining and creates the need for intensive research efforts to develop new effective antimicrobial agents [9,10]. In addition to microorganisms and chemical synthesis, plants are being investigated as new sources of antibacterial and antifungal drugs [11,12,13]. These studies are often focused on the use of whole extracts or fractionated extracts as their antimicrobial activity is a result of the synergistic action of many of their components, whereas the isolation of individual substances is rather expensive and not industrially scalable; furthermore, it often leads to the loss of activity [13].

*Carlina acaulis* L. (Asteraceae) has been used in herbal medicine since antiquity. Many historical sources indicate that the plant was used in the treatment of various diseases caused by microorganisms. These data mainly originate from antiquity as well as the 19th and 20th centuries. Furthermore, the descriptions of treatment of dermatological diseases in medieval and Renaissance sources also suggest the use of *C. acaulis* as an antibiotic agent. Moreover, *C. acaulis* formulations are currently used as antimicrobial agents in ethnomedicine, mainly in the Balkans [14]. This activity may be associated with the presence of many biologically active phytochemicals, e.g., carlina oxide (COx) [15,16,17,18,19,20,21,22,23,24], pentacyclic triterpenes [25], and chlorogenic acids [16,26].

Several reports have presented the results of studies on the antimicrobial activity of various *C. acaulis* formulations (Table 1). They are mainly focused on the essential oil isolated from roots, which contains almost pure COx (90–99%) and is regarded as the main pharmacologically active component of the root [14]. The antibiotic effect of the essential oil was investigated by Stojanović-Radić et al. [27] in studies on *E. coli* ATCC 25922, *S. aureus* ATCC 6538, *Klebsiella pneumoniae* ATCC 10031, *Proteus vulgaris* ATCC 8427, *P. aeruginosa* ATCC 9027, and *C. albicans* ATCC 10231, which showed MIC values ranging from 0.02 to 0.78 µL/mL. In the same study, the authors assessed the bactericidal activity of water, apple cider vinegar, and wine decoctions against *S. aureus* ATCC 6538; however, the MIC values (3.1, 0.7, and 1.5%, respectively) indicated negligible antibacterial activity of these preparations [27]. Another study consisted of an assessment of the antimicrobial and antifungal activity of the hexane extract of *C. acaulis* roots. The results showed the following MIC values: 0.5 and 2.0 mg/mL for methicillin-resistant *S. aureus* (MRSA 1000/93 and MRSA ATCC 10442, respectively), 0.002 mg/mL for vancomycin-resistant *Enterococcus* (VRE ATCC 31299), and 0.25 and 0.5 mg/mL for *C. albicans* ATCC 90028 and *C. glabrata* ATCC MYA 2950, respectively [28]. In addition, the antimicrobial and antifungal activities of *C. acaulis* root oil, pure COx, and a nanoemulsion containing COx were studied by Rosato et al. [18]. These authors demonstrated the outstanding antibacterial and antifungal activity of pure COx and the COx in the nanoemulsion form [18]. Recent studies have also indicated the potential antiviral activity of COx against SARS-CoV-2 [29].

The available literature data provide information on the antibiotic activity of *C. acaulis*. However, they are limited to extremely non-polar (essential oil, hexane extract) or strongly polar (water-, vinegar-, and wine-based extract) fractions. There are no data on the antibiotic activity of extracts containing moderately polar compounds and their fractions. In the present study, a methanol extract from *C. acaulis* roots was prepared and fractionated in order to assess the various fractions in terms of their antimicrobial potential and correlate the activity with their phytoconstituents. To this end, the obtained fractions were evaluated for the antibacterial and antifungal activity against thirteen standard strains of human pathogens, including methicillin-sensitive and methicillin-resistant *S. aureus* strains. Finally, in order to verify their safety, the cytotoxicity of these fractions towards various human cells was assessed. The most active fraction was incorporated in an ointment formulation to provide data for a real-world application of this plant in the treatment of the antibiotic resistance of microorganisms.

## 2. Results

### 2.1. Fractions of Carlina acaulis Methanol Root Extract Contain Chlorogenic Acids

Chlorogenic acids and COx are considered to be the main biologically active substances of the *C. acaulis* root [15,25,30]. Since the antimicrobial activity of COx is already relatively well known, we obtained COx-free fractions in this study. There are many data proving the antimicrobial [31,32,33] and antifungal [34,35] activity of chlorogenic acids; therefore, a quantitative analysis of these metabolites in the obtained fractions was carried out.

The HPLC analysis of the obtained fractions showed the presence of four chlorogenic acids, namely neochlorogenic acid (5-CQA), cryptochlorogenic acid (3-CQA), chlorogenic acid (4-CQA), and 3,5-di-caffeoylquinic acid (3,5-CQA) (Figure 1). The highest content of the acids was detected in fraction C, followed by fractions D and A, while they were not found in fraction B. The dominant compound was 3,5-CQA (c.a. 74% of total chlorogenic acids); however, it was found only in fraction C. The second dominant compound was 4-CQA, whose content was 13.5, 6.94, and 2.30 mg/g in fractions C, D, and A, respectively (Table 2). The content of the other acids (5-CQA and 4-CQA) did not exceed 1 mg/g of the extract. The high concentration of chlorogenic acids in fraction C is undoubtedly the result of the removal of the rich organic matrix since previous studies indicated the content of 3-CQA of approx. 6 mg/g in a crude methanol extract [36].

### 2.2. Fractions of Carlina acaulis Methanol Root Extract Exert Antimicrobial Activity

The antimicrobial (Table 3) and antifungal (Table 4) activity of the obtained fractions was assessed. Fraction A was not only inactive against bacterial strains; in fact, it supported the growth of bacteria. Apart from that, it displayed only very modest antifungal activity (Table 4).

In contrast to fraction A, fractions B and C were characterized by considerable antibacterial (Table 3) and antifungal activity (Table 4) in most of the evaluated strains. Fraction B displayed bacteriostatic and bactericidal activity towards *Bacillus cereus* at doses as low as 0.08 mg/mL. For *Staphylococcus* species, the MIC values ranged from 0.16 to 2.5 mg/mL and the MBC values ranged from 0.6 to 5 mg/mL (Table 3). This fraction also had a fungistatic and fungicidal effect on *Candida* yeasts (MIC from 0.6 to 1.25 mg/mL; MFC from 5 to 10 mg/mL). However, fraction B appeared to have little activity against *Staphylococcus epidermidis*, *Salmonella* Typhimurium, *Escherichia coli*, and *Pseudomonas aeruginosa*, as bactericidal activity was observed at relatively high concentrations (from 10 to more than 20 mg/mL).

Fraction C showed pronounced activity against *B. cereus* (MIC and MBC = 0.16 mg/mL), similar to fraction B. In fact, fraction C displayed bactericidal activity against most of the bacterial strains (MIC from 2.5 to 10 mg/mL and MBC from 5 to 20 mg/mL). The strongest activity was observed towards MRSA ATCC 43300, for which the MIC and MBC were equal to 5 mg/mL (Table 3). It was also shown that fraction C exhibited fungistatic activity (MIC from 2.5 to 5 mg/mL and MFC = 10 mg/mL), as indicated in Table 4.

Fraction D did not affect the bacterial or fungal growth (MIC, MBC, and MFC ≥ 20 mg/mL for all the tested strains).

### 2.3. Cytotoxic Profile and Ecotoxicology

The most active fraction B was collected at a very low yield (<0.1 g), which prevented the analysis of its toxicity and limited its future applicability. Thus, further experiments were conducted using the more abundant and almost equiactive fraction C.

Fraction C displayed significant activity against *B. cereus*, i.e., an etiological factor of foodborne illnesses. Thus, we decided to assess its toxicity towards human enterocytes derived from CaCo2 cells, which constitute a model of intestinal epithelium—tissue affected during *B. cereus* infection. Fraction C showed no significant toxicity towards intestinal epithelium in vitro, similar to vancomycin, a glycopeptide antibiotic used in the management of *B. cereus* infections (Figure 2). The highest dose of fraction C tested here was 200 µg/mL and was higher than the MIC and MBC (160 µg/mL) towards *B. cereus*. 

Fraction C displayed significant activity against several strains of *S. aureus* and *C. albicans*. Since both microorganisms are a common cause of skin infections, we evaluated the cytotoxicity of fraction C against normal human fibroblasts of BJ cell lines (Figure 3A). The viability of the BJ cells was not suppressed by fraction C. In fact, there was a modest but significant increase in cellular growth observed at the dose of 200 µg/mL. COx was used a positive control. This compound significantly inhibited the growth of the BJ fibroblasts at the doses from 25 µg/mL and yielded an IC_50_ of 33.3 µg/mL (Figure 3B). 

To assess the ecotoxicity of fraction C, we exposed the developing *Danio rerio* embryos to the increasing concentrations of fraction C (Figure 3C). Fraction C demonstrated a clear inhibitory trend, although it did not reach significance. By means of extrapolation, the concentration causing the half-maximal lethality (LC_50_) of zebrafish embryos was estimated to be around 213 µg/mL. This value was higher than that observed for COx, which killed all the embryos at the dose as low as 25 µg/mL and yielded an LC_50_ of 5.1 µg/mL (Figure 3D).

### 2.4. Application of C. acaulis Extract in Ointment Formulation

The results of the antibacterial and antifungal activity of the ointment with fraction C are presented in Table 5. The MIC values for all the tested *Staphylococcus* strains ranged from 10 to 80 mg/mL, while the MBC values ranged from 40 to 80 mg/mL. The ratio of MBC to MIC exceeded the value of 4 in none of the discussed cases, which indicates the bactericidal activity of the ointment. Also, for the three strains of yeast tested, the ointment showed fungicidal activity (MFC/MIC from 1 to 2), with MIC values ranging from 20 to 40 mg/mL and with the MFC value of 40 mg/mL.

The toxicity of the ointment was assessed in BJ fibroblasts (Figure 4). A significant toxic effect (drop in viability by 30%) was observed at the dose of 200 mg/mL. No cytotoxicity was observed at the lower doses. The control ointment with mupirocin and clotrimazole exhibited pronounced toxicity as a significant drop in BJ viability was observed at the dose of 120 mg/mL.

## 3. Discussion

This study showed that only fractions B and C from the methanolic extract of the *C. acaulis* root had significant antibacterial and antifungal activity. It seems that the fractions were particularly active against *S. epidermidis* (MSSE), *B. cereus*, and strains of *S. aureus*, including MRSA in comparison with extracts from other plant species, mainly those prepared with the use of ethanol and ethanol–water, i.e., solvents of similar polarity to that of methanol. This activity against the same microorganisms was significantly higher than that of *Crocus alatavicus* Regel & Semen. ethanolic extracts, as demonstrated by Allambergenova et al. [37]. For example, the MIC values in the case of *S. aureus* ATCC 25923, ATCC BAA-1707, *S. epidermidis* ATCC 12228, and *B. cereus* ATCC 10876 strains were 20 mg/mL [37]. Thus, the bacteriostatic activity of fraction B was about 33 and 67 times higher for *S. aureus* ATCC 25923 and ATCC BAA 1707, respectively, 2-fold higher for *S. epidermalis* ATCC 12228, and as much as 250 times higher for *B. cereus* ATCC 10876. Also, fraction C had eight times higher bacteriostatic activity for *S. aureus* ATCC 25923, 2-fold higher for *S. aureus* ATCC BAA 1707, four times higher for *S. epidermidis* ATCC 12228, and 125 times higher for *B. cereus* ATCC 10876. The bacteriostatic and bactericidal activity against the *B. cereus* of both tested fractions (MIC = 0.08 and MBC = 0.16 mg/mL for fractions B and C, respectively) seems to be particularly important in light of the studies of extracts from *Artemisia gmelinii*, which showed inhibitory activity against *Bacillus* spp. at concentrations ranging from 1.25 to 5 mg/mL [38]. Interestingly, both fraction B and fraction C had bactericidal effects on *B. cereus* ATCC 10876 at very low concentrations, identical to the bacteriostatic concentrations. These observations make the obtained fractions a promising target for future studies of antimicrobial activity against *B. cereus.*

Fractions B and C were also characterized by the strongest antifungal activity. The inhibitory activity of fraction B against *C. albicans* ATCC 10231 was shown to be more than sixteen times greater than that reported by Allambergenova et al. [37], who used extracts from *C. alatavicus*. The growth of *C. glabrata* ATCC 90030 was also inhibited at 8-fold lower concentrations of fraction B than those noted by Allambergenova et al. [37]. What is noteworthy is the comparison of the antifungal activity of fractions B and C with the results obtained by Malm et al. [39] in a study on extracts from *Helianthus salicifolius* A. Dietr. and *H. tuberosus* L. which revealed that fraction B showed more than 8-fold stronger inhibitory activity against *C. albicans* ATCC 10231 and *C. glabrata* ATCC 90030, while fraction C inhibited the growth of these yeasts at two-times lower concentrations than *Helianthus* extracts. Fraction B also exerted a stronger inhibitory effect on *Candida* yeasts compared to extracts from *A. gmelinii*, while the MIC values of fraction C against *Candida* spp. were the same as those obtained by Mamatova et al. [38]. It should be noted that fraction C had a fungistatic effect when applied at higher concentrations than fraction B, but its fungicidal activity was demonstrated against all yeasts tested in contrast to fraction B, which had a fungicidal effect only on *C. glabrata*.

Extracts from plants used in ethnomedicine are routinely screened for antibacterial and antifungal activities [40,41,42]. For instance, the hydroalcoholic extract of *Lycium shawii* showed some antibacterial activity against *B. cereus* ATCC 10876 with an MIC value of 12.5 mg/mL [43], whereas the ethanolic stem bark extract of *Clausena heptaphylla* yielded an MIC value of 2.5 mg/mL [44]. Fraction B (heptane) and C (ethyl acetate) of the methanol extract from *C. acaulis* described in this study were substantially more active against the same bacterial strain (MIC equal to 0.08 and 0.16, respectively) and only slightly less active in comparison with the essential oil’s isolated form *Thymus willdenowii* Boiss & Reut (MIC values ranging from 0.01 to 0.03 mg/mL, depending on the part of the plant) [45]. Fractions B and C reported here displayed bactericidal activity against *S. aureus* ATCC 43300 with MIC values of 2.5 and 5 mg/mL, respectively. There are some identified materials derived from other plants that display more pronounced activity against this bacterial strain, especially the n-hexane fraction of the methanol extract of the stem bark of *Chrysophyllum lacourtianum* (MIC = 0.13 mg/mL) [46], the pinocembrin-7-O residue-rich fraction from *Penthorum chinense* Pursh stems (MIC = 0.06 mg/mL), and ethanolic extracts of *Erodium gruinum*, *Euphorbia hierosolymitana*, and *Tamarix tetragyna* (MIC = 0.001 mg/mL in all three cases) [47]. Apart from antibacterial activities, the fractions obtained in this study also displayed some antifungal properties. For example, fraction B was fungicidal against *C. glabrata* ATCC 90030 with an MIC of 1.25 mg/mL. This is comparable to the crude ethanolic extract obtained from the leaves of *Commiphora leptophloeos* (Mart.) J.B. Gillett (Burseraceae) which yielded an MIC equal to 1.02 mg/mL in the same yeast strain [48]. However, methanol extracts from the fruit pulp and seeds of *Cassia fistula* Linn. (Caesalpiniaceae) were more active with MIC values of 0.10 and 0.30 mg/mL, respectively [49].

The antimicrobial activity of chlorogenic acid has been demonstrated in numerous studies [31,32,33,34,50,51,52]. Lou et al. showed that the MIC values of pure chlorogenic acid against six bacterial strains (*Streptococcus pneumoniae*, *Bacillus subtilis*, *Staphylococcus aureus*, *Shigella dysenteriae*, *Escherichia coli*, and *Salmonella* Typhimurium ranged from 20 to 80 µg/mL [31]. Our study shows that the MIC values of fraction C are in the range of 2.5–10 mg/mL for most microorganisms (with the exception of *B. cereus*). Thus, if the chlorogenic acid content of fraction C is 13.5 mg/g (Table 2), the concentration of this compound in solutions that inhibit the growth of microorganisms is in the range of 33.75 to 135 µg/mL (for *B. cereus* 2.16 ug/mL). A comparison of the results obtained by Lou et al. for pure chlorogenic acid [31] and the concentrations of this compound in microbial inhibitory solutions in our experiment significantly explains the antimicrobial activity of the C fraction. Lou et al. also found that chlorogenic acid increases the permeability of the cell membrane, causing cytoplasm leakage and the depletion of intracellular potential, resulting in cell death [31]. They observed these processes using transmission electron microscopy, which very strongly explains the mechanism of chlorogenic acid antimicrobial activity. Han et al. showed that the MIC of 3,5-CQA against *Bacillus shigae* was 160 µg/mL [50]. Higher concentrations of this acid were achieved in our experiment in solutions with MICs of 5 and 10 mg/mL of the C fraction (189.45 and 378.9 µg/mL of 3,5-CQA, respectively). However, there are also data indicating that *S. aureus* strains are significantly resistant to chlorogenic acid. Li et al. showed that for eight strains of this pathogen, the MIC values of chlorogenic acid ranged from 2.5 to 5 mg/mL [51]. These values were not reached in our experiment, even taking into account the sum of chlorogenic acids contained in the C fraction. Thus, the literature data indicate a very large variation in the sensitivity of pathogens to chlorogenic acids, so comparing them to the results we obtained does not allow us to draw clear conclusions, especially since fraction C contains a composition of chlorogenic acids.

A comparison of the present data with those presented by Rosato et al. [18] for the antibacterial and antifungal activity of *C. acaulis* root oil and pure COx may indicate that the oil has comparable antibacterial activity against *S. aureus* to that of fraction C (similar values of the MIC parameter), while pure COx shows similar activity to that of fraction B. Interestingly, fractions B and C are much more active against *B. cereus* than the oil and pure COx, while these lipophilic components of *C. acaulis* show very strong antifungal activity in contrast to the fractions from the methanolic extract. The fact that fractions B and C show strong antimicrobial activity indicates that COx is not the only metabolite of *C. acaulis* with such activity, and the use of this plant in traditional medicine should not be attributed to this substance alone.

Although fraction B was characterized by strong antimicrobial activity (especially against *B. cereus*), its amount obtainable from the roots of *Carlina* plants seems too small to be used in future pharmaceutical preparations. Moreover, this fraction requires additional phytochemical characterization, as no compounds have been determined in its composition to date. For this reason, fraction C, which can be obtained in large quantities and is rich in chlorogenic acids, was chosen for the preparation of an experimental medicinal preparation.

Although the MIC and MBC/MFC values of the fraction C ointment against the tested strains of microorganisms were high (compared to the ointment with mupirocin and clotrimazole), it should be noted that the MBC/MIC and MFC/MIC ratio for the fraction C ointment did not exceed 4 (Table 5). This demonstrates the bactericidal and fungicidal activity of the tested ointment. Analogous parameters for ointments with mupirocin and clotrimazole were significantly higher, indicating their bacteriostatic and fungistatic effects. Moreover, the toxicity of the ointment with mupirocin and clotrimazole to fibroblasts at concentrations as low as 120 mg/mL was demonstrated, while the ointment containing fraction C was not toxic, even at a concentration of 160 mg/mL (Figure 4), which may indicate the safety of the developed formulation. Although the concentration range of MBC and MFC for mupirocin and clotrimazole ointments is much lower than the toxic concentration range, it should be remembered that the dosage of ointments is not as precise as the dosage of the drug, and for this reason, one should strive for the lowest toxicity of this type of preparation.

The MIC values for Hascobase ranged from 20 to160 mg/mL and from 40 to 80 mg/mL for bacteria and fungi, respectively. The MBC values for Hascobase ranged from 160 to >320 mg/mL and 80–320 mg/mL for bacteria and fungi, respectively. Therefore, these values were higher than the activity of the ointment with fraction C. It should be noted that in most cases, the activity of Hascobase showed bacteriostatic/fungistatic activity (MBC/MIC > 8 and MFC/MIC > 8) or could not be determined (MBC and MFC >320 mg/mL) (Table 5), while the ointment with fraction C was showed lower MIC values with bactericidal/fungicidal activity.

## 4. Materials and Methods

### 4.1. Reference Standards and Chemicals

Chlorogenic acid (≥95%), neochlorogenic acid (≥98%), cryptochlorogenic acid (≥98%), 3,5-di-caffeoylquinic acid (≥95%), trifluoroacetic acid (≥99%), and HPLC-grade methanol and acetonitrile were purchased from Sigma Aldrich (St. Louis, MO, USA). Water for HPLC was purified by Ultrapure Milli-pore Direct-Q^®^3UV–R (Merck Millipore, Billerica, MA, USA). The COx with 96.2% purity was obtained by the distillation of *C. acaulis* roots in the Deryng apparatus. The identity and purity of the compound were confirmed in accordance with previously established methodology [30]. The ointment with mupirocin (Mupirox, Pharmaswiss Ceska Republika, Prague, Czech Republic) and clotrimazole (Clotrimazolum Hasco, Hasco-Lek S.A., Wroclaw, Poland) was purchased from a local pharmacy.

### 4.2. Plant Material, Preparation and Fractionation of the Extract

The *C. acaulis* (voucher specimen no. 684) plants were obtained from the Botanical Garden of Maria Curie-Skłodowska University in Lublin (latitude 51°16′ N, longitude 22°30′ E, altitude: 178–217 m a.s.l.). The plants were collected in the second half of July 2020. The roots were thoroughly washed with tap and distilled water, dried at room temperature, and pulverized. Exactly 300 g of roots were extracted four times (4 × 30 min) with methanol (4 × 3 L) using an ultrasonic bath (Bandelin Sonorex RK 510 H, BANDELIN electronic GmbH & Co. KG, Berlin, Germany) with a frequency of 35 kHz. The obtained extracts were combined and evaporated to 1 L, using a rotary evaporator (Heidolph Hei-Vap Expert, Heidolph Instruments GmbH & Co. KG, Schwabach, Germany) under a pressure of 150 mbar and at 50 °C. The precipitate isolated from the concentrated extract was filtered through a filter paper (fraction A—1.4 g). The remaining extract was diluted 1:1 (*v*/*v*) with deionized water and extracted with heptane in a separatory funnel (6 × 100 mL of heptane). The heptane fraction (containing COx) was evaporated on a rotary evaporator (90 mbar; 50 °C), frozen at -80 °C (freezer Eppendorf CryoCube F440n, Hamburg, Germany), and dried by lyophilization (freeze-dryer Christ Alpha 2–4 LDplus, Christ, Osterode am Harz, Germany; 0.001 mbar for 7 days) to yield COx-free fraction B (<0.1 g). Next, the methanol–water mixture was extracted with ethyl acetate (6 × 100 mL). The resulting acetate fraction was evaporated to dryness using a rotary evaporator (Heidolph Instruments, Schwabach, Germany) (145 mbar; 50 °C), and the process yielded 5.76 g of a dry residue (fraction C). The remaining methanol–water fraction was concentrated on a rotary evaporator (150 mbar; 50 °C), frozen at −80 °C, and then dried by lyophilization (0.001 mbar for 7 days) to yield fraction D (45.51 g). The fractionation procedure is shown in Figure 5.

### 4.3. HPLC-PDA Analysis of Chlorogenic Acids

The following analytical conditions and devices were used: an EliteLaChrom chromatograph with a PDA detector and EZChrom Elite software version 3.3.2 (Merck, Darmstadt, Germany), a C18 reversed-phase core–shell column (Kinetex, Phenomenex, Aschaffenburg, Germany) (25 cm × 4.6 mm i.d., 5 µm particle size) at 25 °C, a mixture of water with 0.025% of trifluoroacetic acid (solvent A), and acetonitrile with 0.025% of trifluoroacetic acid (solvent B). Gradient elution: 0.0–5.0 min 95% A, 5% B; 5.0–60 min A from 5% to 20%, and B from 95% to 80%. The flow rate was 1.0 mL/min. Data were collected between 190 and 450 nm. The identity of the compounds was established via a comparison of the retention times and PDA spectra with the corresponding standards. Quantitative analysis was performed at λ = 324 nm. 

### 4.4. Antimicrobial Activity Assessment

The antimicrobial profile of the fractions obtained from the *C. acaulis* root extract was evaluated in a panel of microorganisms from the American Type Culture Collection (ATCC), including Gram-positive bacteria (*Staphylococcus aureus* ATCC 25923, *Staphylococcus aureus* ATCC 6538, *Staphylococcus aureus* ATCC 29213, *Staphylococcus aureus* ATCC BAA 1707, *Staphylococcus aureus* ATCC 43300, *Staphylococcus epidermidis* ATCC 12228, and *Bacillus cereus* ATCC 10876), Gram-negative bacteria (*Salmonella* Typhimurium ATCC 14028, *Escherichia coli* ATCC 25922, and *Pseudomonas aeruginosa* ATCC 27853), and yeasts (*Candida albicans* ATCC 10231, *Candida glabrata* ATCC 90030, and *Candida krusei* ATCC 14243). The strains were provided by the local collection of the Department of Pharmaceutical Microbiology, Medical University in Lublin. Microbial suspensions were prepared in sterile saline (0.85% NaCl) with an optical density of 0.5 McFarland standard—1.5 × 10^8^ colony-forming units (CFUs) per mL. Mueller–Hinton medium (Biomaxima, Lublin, Poland) was used with a series of 2-fold dilutions of the tested substances in a range of final concentrations from 80 to 0.08 mg/mL. The in vitro antibacterial activity of all tested compounds was screened on the basis of the minimal inhibitory concentration (MIC). It was determined with the broth microdilution method, based on the European Committee on Antimicrobial Susceptibility Testing (EUCAST) guidelines [53]. MBC (minimal bactericidal concentration) and MFC (minimal fungicidal concentration) were estimated with the broth microdilution technique by plating out the contents of wells that showed no visible growth of bacteria onto Mueller–Hinton agar and incubating at 35 °C for 18 h. The MIC, MBC, and MFC values were given in mg/mL in accordance with the EUCAST references [53].

### 4.5. Cell Culture

Human BJ foreskin fibroblasts (RRID:CVCL_3653; [54]) were obtained from the ATCC (#CRL-2522). The cells were maintained in high glucose (4.5 g/L) Dulbecco’s modified Eagle’s medium (DMEM) supplemented with sodium pyruvate (1 mmol), penicillin (100 U/mL), streptomycin (0.1 mg/mL), and fetal bovine serum (FBS, 10%), all from ThermoFisher Scientific (Waltham, MA, USA). 

Human enterocytes were obtained by the differentiation of colorectal adenocarcinoma CaCo2 cells (RRID:CVCL_0025; ATCC, #HTB-37; [55]). The cells were routinely maintained in Eagle’s minimum essential medium (EMEM) supplemented with penicillin (100 U/mL), streptomycin (0.1 mg/mL), and FBS (10%). The differentiation into polarized enterocytes was carried out on 96-well plates, where CaCo2 cells were seeded out and maintained in the culture for two weeks. The medium was replaced every two days. Ultimately, a sealed monolayer of CaCo2-derived enterocytes was obtained [56].

All cells were cultured in an incubator maintaining a humidified atmosphere of 95% air and 5% CO_2_ and a temperature of 37 °C.

### 4.6. Cytotoxicity Assay

The toxicity of the *C*. *acaulis* root fractions was assessed with the MTT assay [57]. The fibroblasts were seeded out onto a 96-well plate at 12 × 10^3^ cells/well in 100 µL of full culture medium. The cells were allowed to attach overnight. The next day, the medium was replaced with a fresh one containing decreasing concentrations of the extracts (200, 100, 50, 25, 12.5, 6.25, 3.13, and 1.56 µg/mL) or vehicle (DMSO, 0.5%). Alternatively, the cells were exposed to the ointment dissolved directly in full cell culture medium at the desired concentration. The differentiated CaCo2 cells were subjected to the same treatment regime.

The cells were treated for 24 h. Then, 10 µL of MTT solution (5 mg/mL) in phosphate-buffered saline (PBS) was added to each well. The plates with MTT were incubated inside a cell culture incubator (37 °C) for 3 h. Subsequently, the medium was removed from above the cells by gentle aspiration. The remaining formazan crystals were dissolved in DMSO (100 µL/well). After 3 min of agitation, the plates were inserted into an Epoch plate reader (Biotek Instruments, Winooski, VT, USA), and the absorbance was recorded at 560 nm and 620 nm (reference wavelength).

### 4.7. Ecotoxicity Assessment

The eggs of zebrafish (*Danio rerio*; AB line) were obtained from the Centre of Experimental Medicine, Medical University of Lublin, Poland. The zebrafish embryo acute toxicity (ZFET) procedure was based on the OECD Guidelines for the Testing of Chemicals, Test No. 236 [58], adapted by Nishimura et al. [59]. In brief, 20 to 24 embryos per concentration were exposed to the extracts (1.56, 3.13, 6.25, 12.5, 25, 50, 100, and 200 μg/mL) or 0.5% DMSO (solvent control). COx derived from *C. acaulis* was used as a positive control as this natural compound had been previously determined to be toxic to zebrafish embryos [17]. The exposure was initiated at the 16-cell stage and was continued for 96 h. The embryos (five to six embryos per well) were distributed across a 24-well plate and maintained at 28.5 ± 0.5 °C with a day–night cycle (14 h light/10 h dark). Acute toxicity was assessed every 24 h by counting the occurrence of the following indicators of lethality: embryo coagulation, the lack of somite formation, the non-detachment of the tail, and the lack of a heartbeat. At the end of the test, acute toxicity was determined based on the percentage of dead zebrafish.

### 4.8. Preparation of the Ointment

Fraction C was dissolved in a mixture of ethanol and DMSO (8:10 *v*/*v*) in the ratio of 5.6:94.4 (*m*/*v*). The resulting solution was emulsified in a commercially available amphiphilic ointment base Hascobaza (Hasco-Lek S.A., Wrocław, Poland) composed of liquid paraffin 3.0 g, white Vaseline 32.0 g, glyceryl monostearate 3.0 g, cetostearate alcohol 9.0 g, Tween 40^®^ 7.0 g, Miglyol 812^®^ 2.0 g, propylene glycol 5.0 g, colloidal anhydrous sillica 0.1 g, sorbic acid 0.2 g, purified water to 100 g, with the addition of glycerol in the following weight proportions: 19:4:77 of fraction C solution, glycerol, and base, respectively. The concentration of fraction C in the obtained ointment was 1%, while that of ethanol, DMSO, and glycerol was 8, 10, and 4%, respectively.

### 4.9. Data Analysis

The half-maximal inhibitory concentration (IC_50_) and median lethal dose (LC_50_) were determined by nonlinear four-parameter regression analysis. Dose–response curves were generated using Prism 8.4.3 (GraphPad Software, San Diego, CA, USA). Bartlett’s test was used to test for equal variances. Brown–Forsythe ANOVA with Dunnett’s T3 post hoc test were used to compare the mean viability of the cells exposed to the different concentrations of fraction C or COx to the mean viability of vehicle-treated cells. Two-way ANOVA followed by Tukey’s multiple comparisons test were used to compare the effects of two factors (i.e., ointment composition and dose) on the viability of cultured cells.

## 5. Conclusions

Our studies have shown that the fractionation of the methanolic extract of *C. acaulis* roots can yield fractions with potent antibacterial and antifungal properties, especially against *B. cereus* and methicillin-resistant *S. aureus.* The presence of a large amount of chlorogenic acids in the active fraction largely explains the observed biological effects. The methanol extract fractionation methodology may be useful for the industrial isolation of pure chlorogenic acids, of which *C. acaulis* is a very rich source. In addition, the fraction rich in chlorogenic acids can be used to produce ointments with antimicrobial activity. This may explain the use of *C. acaulis* root preparations in ethnomedicine as antimicrobial drugs. 

## 6. Patents

The following patent applications were filed to the Polish Patent Office as a result of the research: (1) “Method for obtaining an antiseptic and fungicidal preparation and the antiseptic and fungicidal preparation”, application number P.440027, and (2) “Method for obtaining the bactericidal fraction of the extract from the root of the stemless carline thistle (*Carlina acaulis* L.)”, application number P.440022.

## Figures and Tables

**Figure 1 molecules-29-01939-f001:**
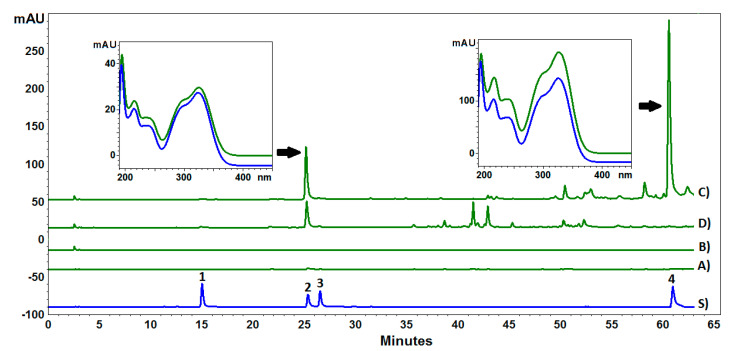
HPLC-PDA chromatograms of methanolic extract fractions from *Carlina acaulis* root (A–D) and a mixture of reference standards (S): 1—neochlorogenic acid, 2—chlorogenic acid, 3—cryptochlorogenic acid, 4—3,5-di-caffeoylquinic acid.

**Figure 2 molecules-29-01939-f002:**
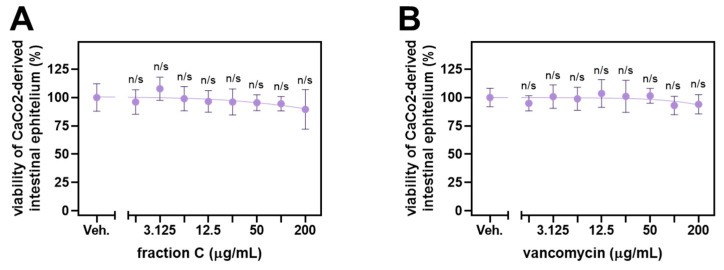
Viability of human enterocytes-based monolayer obtained through differentiation of CaCo2 cells upon 24 h exposure to fraction C (**A**) or vancomycin (**B**). Error bars represent standard deviation. The statistical analysis involved Brown–Forsythe ANOVA with Dunnett’s T3 post hoc test. n/s, not significant vs. vehicle control. A four-parameter dose–response curve was fitted to the experimental data points; however, it was not possible to determine the inhibitory concentration causing a half-maximal drop in cellular viability (IC_50_).

**Figure 3 molecules-29-01939-f003:**
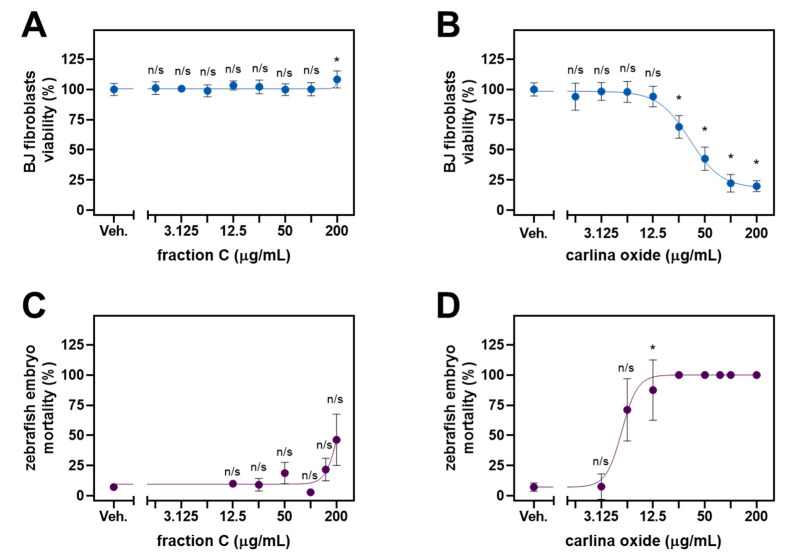
Viability of normal human BJ fibroblasts and zebrafish embryos treated with fraction C or carlina oxide (COx). The BJ cells were exposed to either fraction C (**A**) or COx (**B**) for 24 h, and the fitness of the cells was assessed by the MTT assay. Ecotoxic activity was assessed in zebrafish embryos developing in the presence of either fraction C (**C**) or COx (**D**) for 96 h. Error bars represent standard deviation. The statistical analysis involved Brown–Forsythe ANOVA with Dunnett’s T3 post hoc test. n/s, not significant vs. vehicle control; * *p* < 0.05.

**Figure 4 molecules-29-01939-f004:**
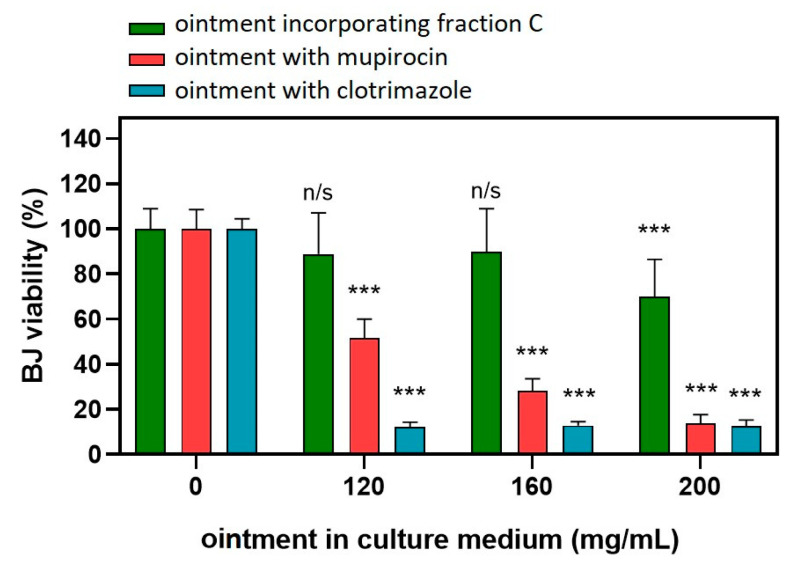
Effect of ointment incorporating fraction C (green bars) or with mupirocin (red bars) on the viability of BJ fibroblasts. Statistical analysis: two-way ANOVA followed by Tukey’s multiple comparisons test. n/s, not significant vs. respective vehicle control; *** *p* < 0.001.

**Figure 5 molecules-29-01939-f005:**
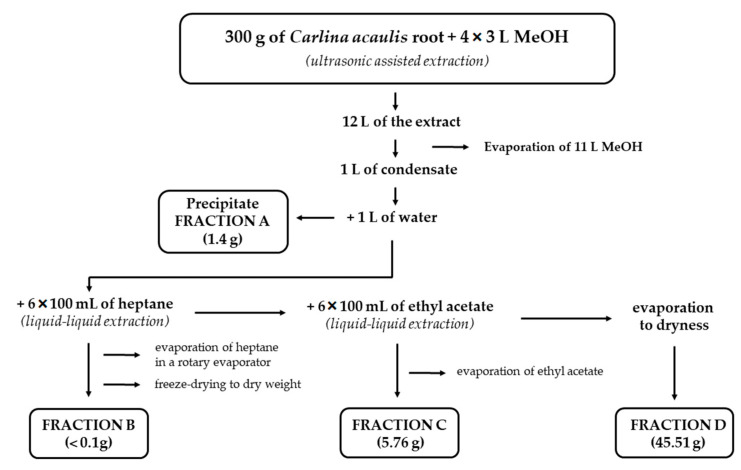
Procedure for fractionation of the methanol extract from *Carlina acaulis* L. root.

**Table 1 molecules-29-01939-t001:** Antimicrobial activities of *C. acaulis*-based formulations described in the literature.

Species	Extract Type	Activity	Reference
*Escherichia coli* ATCC 8739	Essential oil	MIC 0.39 µL/mL	[27]
*E. coli* ATCC 25922	Essential oil	MIC 0.78 µL/mL	[27]
*Staphylococcus aureus* ATCC 6538	Essential oil	MIC 0.02 µL/mL	[27]
*Klebsiella pneumoniae* ATCC 10031	Essential oil	MIC 0.78 µL/mL	[27]
*Proteus vulgaris* ATCC 8427	Essential oil	MIC 0.09 µL/mL	[27]
*Pseudomonas aeruginosa* ATCC 9027	Essential oil	MIC 0.09 µL/mL	[27]
*Candida albicans* ATCC 10231	Essential oil	MIC 0.19 µL/mL	[27]
*S. aureus* ATCC 6538	Water decoction	MIC 3.1%	[27]
*S. aureus* ATCC 6538	Apple cider vinegar decoction	MIC 0.7%	[27]
*S. aureus* ATCC 6538	Wine decoction	MIC 1.5%	[27]
MRSA 1000/93	Hexane extract	MIC 0.5 mg/mL	[28]
MRSA ATCC 10442	Hexane extract	MIC 2.0 mg/mL	[28]
VRE ATCC 31299	Hexane extract	MIC 0.002 mg/mL	[28]
*C. albicans* ATCC 90028	Hexane extract	MIC 0.25 mg/mL	[28]
*C. glabrata* ATCC MYA 2950	Hexane extract	MIC 0.5 mg/mL	[28]
*C. albicans* ATCC 10231	Essential oil	0.68 mg/mL	[28]
*C. albicans* ATCC 10231	COx	0.04 mg/mL	[28]
*C. albicans* ATCC 10231	COx nanoemulsion	1.9 mg/mL	[28]
SARS-CoV-2	COx	IC50 = 234.2 µg/mL in binding to host cell receptor	[29]

MRSA—methicillin-resistant *Staphylococcus aureus*; VRE—vancomycin-resistant *Enterococcus*; SARS-CoV-2—severe acute respiratory syndrome coronavirus 2.

**Table 2 molecules-29-01939-t002:** Contents of chlorogenic acids (mg/g fraction ± SD) in fractions obtained from *Carlina acaulis* L. root. N = 3.

Fraction	5-CQA	3-CQA	4-CQA	3,5-CQA	Total Content
A	n.d.	2.30 ± 0.09	0.40 ± 0.02	n.d.	2.69
B	n.d.	n.d.	n.d.	n.d.	0.00
C	0.12 ± 0.04	13.15 ± 0.39	0.68 ± 0.07	37.89 ± 1.75	51.84
D	0.25 ± 0.03	6.94 ± 0.04	0.81 ± 0.01	n.d.	8.00

5-CQA, neochlorogenic acid; 3-CQA, chlorogenic acid; 4-CQA, cryptochlorogenic acid; 3,5-CQA, 3,5-di-caffeoylquinic acid; n.d., not detected.

**Table 3 molecules-29-01939-t003:** Antibacterial activity of *Carlina acaulis* methanol extract fractions.

	Fraction B			Fraction C			Fraction D		
	MIC	MBC	MBC/MIC	MIC	MBC	MBC/MIC	MIC	MBC	MBC/MIC
	mg/mL			mg/mL			mg/mL		
*Staphylococcus aureus* ATCC 25923 (MSSA)	0.6	2.5	4 bactericidal	2.5	20	8 bacteriostatic	20	>20	none
*Staphylococcus aureus* ATCC 6538 (MSSA)	0.16	0.6	4 bactericidal	2.5	20	8 bacteriostatic	20	>20	none
*Staphylococcus aureus* ATCC 29213 (MSSA)	0.6	5	8 bacteriostatic	5	10	2 bactericidal	20	>20	none
*Staphylococcus aureus* ATCC BAA 1707 (MRSA)	0.3	2.5	8 bacteriostatic	10	10	1 bactericidal	>20	>20	none
*Staphylococcus aureus* ATCC 43300 (MRSA)	2.5	5	2 bactericidal	5	5	1 bactericidal	20	>20	none
*Staphylococcus epidermidis* ATCC 12228 (MSSE)	10	10	1 bactericidal	5	20	4 bactericidal	20	>20	none
*Bacillus cereus* ATCC 10876	0.08	0.08	1 bactericidal	0.16	0.16	1 bactericidal	20	20	1 bactericidal
*Salmonella* Typhimurium ATCC 14028	10	>20	bactericidal	10	20	2 bactericidal	20	>20	none
*Escherichia coli* ATCC 25922	20	20	1 bactericidal	10	20	2 bactericidal	20	>20	none
*Pseudomonas aeruginosa* ATCC 27853	5	20	4 bactericidal	10	20	2 bactericidal	20	>20	none

MIC—minimal inhibitory concentration; MBC—minimal bactericidal concentration; MSSA—methicillin-sensitive *Staphylococcus aureus*; MRSA—methicillin-resistant *Staphylococcus aureus*; MSSE—methicillin-sensitive *Staphylococcus epidermidis*.

**Table 4 molecules-29-01939-t004:** Antifungal activity of *Carlina acaulis* methanol extract fractions.

	Fraction A			Fraction B			Fraction C			Fraction D		
	MIC	MFC	MFC/MIC	MIC	MFC	MFC/MIC	MIC	MFC	MFC/MIC	MIC	MFC	MFC/MIC
	mg/mL			mg/mL			mg/mL			mg/mL		
*Candida albicans* ATCC 10231	10	20	2 fungicidal	0.6	10	16 fungistatic	2.5	10	4 fungicidal	>20	>20	- none
*Candida glabrata* ATCC 90030	20	20	1 fungicidal	1.25	5	4 fungicidal	5	10	2 fungicidal	>20	>20	- none
*Candida krusei* ATCC 14243	10	20	2 fungicidal	1.25	10	8 fungistatic	5	10	2 fungicidal	>20	>20	- none

MIC—minimal inhibitory concentration; MFC—minimal fungicidal concentration.

**Table 5 molecules-29-01939-t005:** Antibacterial and antifungal activity of the ointment incorporating fraction C.

	Ointment Base (Hascobaza)			Ointment with Fraction C			Ointment with Mupirocin (2%)		
	MIC	MBC	MBC/MIC	MIC	MBC	MBC/MIC	MIC	MBC	MBC/MIC
	mg/mL			mg/mL			mg/mL		
*Staphylococcus aureus* ATCC 25923 (MSSA)	80	>320	none	40	40	1	0.78	12.5	16
*Staphylococcus aureus* ATCC 6538 (MSSA)	40	320	8	20	80	4	0.39	12.5	32
*Staphylococcus aureus* ATCC 29213 (MSSA)	80	320	4	20	80	4	0.39	25	64
*Staphylococcus aureus* ATCC BAA 1707 (MRSA)	80	>320	none	40	80	2	0.78	12.5	16
*Staphylococcus aureus* ATCC 43300 (MRSA)	160	>320	none	80	80	1	0.39	12.5	32
*Staphylococcus epidermidis* ATCC 12228 (MSSE)	20	160	8	10	40	4	0.39	12.5	32
	**Ointment Base** **(Hascobaza)**			**Ointment with Fraction C**			**Ointment with** **Clotrimazole (1%)**		
	**MIC**	**MFC**	**MFC/MIC**	**MIC**	**MFC**	**MFC/MIC**	**MIC**	**MFC**	**MFC/MIC**
	**mg/mL**			**mg/mL**			**mg/mL**		
*Candida albicans* ATCC 10231	40	80	2	20	40	2	0.03	0.5	16
*Candida glabrata* ATCC 90030	80	160	2	40	40	1	0.03	0.5	16
*Candida krusei* ATCC 14243	40	320	8	20	40	2	0.002	0.03	16

MIC, minimal inhibitory concentration; MBC, minimal bactericidal concentration; MFC, minimal fungicidal concentration; MSSA, methicillin-sensitive *Staphylococcus aureus*; MRSA, methicillin-resistant *Staphylococcus aureus;* MSSE, methicillin-sensitive *Staphylococcus epidermidis.*

## Data Availability

All data are contained within the article. Raw datapoints from the study are available on request from the corresponding author.

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
