# Peer review of "Fractionation of Carlina acaulis L. Root Methanolic Extract as a Promising Path towards New Formulations against Bacillus cereus and Methicillin-Resistant Staphylococcus aureus"

_molecules, 2024, doi:10.3390/molecules29091939_

Round 1

Reviewer 1 Report

Comments and Suggestions for Authors

The compound 4 the main component of the fraction C might be a bought in market https://www.sigmaaldrich.com/RU/en/product/sigma/smb00131. What reason to isolate this compound from natural object?

Line 110-115 it is not results.

Line 133. Please add the «HPLC-PDA». It is clearer for readers

Line 127. Table 1. How was the mass of components in fractions calculated?

Line 156-157. Italics

Table 4 look clearer than tables 2 and 3.

In general, I don't see any serious comments for an article of this level. However, I would like to note that the study of the biological activity of the extract is inferior to the study of the biological activity of individual compounds with an established chemical structure by spectral methods including NMR and MS. Taking into account the described methods for identifying the components of fractions and comparing their chromatograms with standards using the IK, it can be considered reasonable, but not full complete.

Author Response

R1. Q1. The compound 4 the main component of the fraction C might be a bought in market https://www.sigmaaldrich.com/RU/en/product/sigma/smb00131. What reason to isolate this compound from natural object?

Response to R1. Q1. While chlorogenic acids are commercially available from multiple vendors, their most common source is actually isolation from plant material. This highlights the importance of both identification of new plant sources and development of efficient isolation methods for chlorogenic acids. It is worth noting that during our research, the role of chlorogenic acids in the observed antimicrobial activity was not initially suspected.  The parallel investigations of biological activity and chemical composition yielded promising results, that suggested potential value in both phytotherapy applications and cost-effective isolation of pure chlorogenic acids.

R1. Q2. Line 110-115 it is not results.

Response to R1. Q2. We agree with the reviewer that these sentences do not directly present results. We believe that these three sentences provide a concise justification for the research undertaken, aiding a smooth transition from the introduction to the results section.  We would propose keeping them in their current position.

R1. Q3. Line 133. Please add the «HPLC-PDA». It is clearer for readers

Response to R1. Q3. The indication of PDA detection usage was added as requested.

R1. Q4. Line 127. Table 1. How was the mass of components in fractions calculated?

Response to R1. Q4. A standard procedure for quantitative analysis was used. Accurately weighed samples (ok. 10 mg) of each fraction were dissolved in methanol (10 mL) and analyzed by HPLC-PDA. The content of analytes was calculated from calibration curves made by injecting a series of standard solutions and converted to the dry weight of the fractions.

R1. Q5. Line 156-157. Italics

Response to R1. Q5. The formatting of the lattin names was changed to italics.

R1. Q6. Table 4 look clearer than tables 2 and 3.

Response to R1. Q6. Original Tables 2 and 3 (new Tables 3 and 4) were redesigned for increased clarity.

R1. Q7. In general, I don't see any serious comments for an article of this level. However, I would like to note that the study of the biological activity of the extract is inferior to the study of the biological activity of individual compounds with an established chemical structure by spectral methods including NMR and MS. Taking into account the described methods for identifying the components of fractions and comparing their chromatograms with standards using the IK, it can be considered reasonable, but not full complete.

Response to R1. Q7. We agree with the reviewer and will determine the activity of pure compounds in the future. For the identification and quantification of chlorogenic acids, we relied on well-documented data showing that chlorogenic acids are among the main metabolites of C. acaulis (there are many publications with MS data). In such a situation, we felt that comparing retention times and PDA spectra of peaks of standard substances and peaks from chromatograms of samples was sufficient.

Reviewer 2 Report

Comments and Suggestions for Authors

The manuscript "Fractionation of Carlina acaulis L. root methanolic extract as a promising path towards new formulations against Bacillus cereus and methicillin-resistant Staphylococcus aureus" is interesting, but the validity of using methanolic extracts is questionable.

Line 29: full name of the plant

Line 53: Incorrect citation

Lines 135-139: has already been written in the methodology

Line 141: Table 2 not 1

Line 157-158: Latin names of microorganisms - italics

Line 228: „p” italics

A discussion based on 4 references is very modest

Line 301: full name

Line 316: methanol which concentration

Line 316: “ultrasonic bath” detailed equipment data

Line 317: „evaporated” detailed equipment data

Line 318: there are many kinds of filters

Line 321: „lyophilization” detailed equipment data

Fig 5. hardly legible

Line 366: currently "mol" not "M"

From Line 367: mL

Author Response

R2. Q1. The manuscript "Fractionation of Carlina acaulis L. root methanolic extract as a promising path towards new formulations against Bacillus cereus and methicillin-resistant Staphylococcus aureus" is interesting, but the validity of using methanolic extracts is questionable.

Response to R2. Q1. Dear Reviewer we, too, hoped for stronger antimicrobial activity of the obtained fractions and the obtained results are relatively unsatisfactory, especially compared to the activity of antibiotics. However, we believe that the results of the conducted research are valuable at least in terms of expanding the state of knowledge about C. acaulis which is used in ethnomedicine as an antimicrobial agent. In addition, this research contributes to the development of procedures for the isolation of pure chlorogenic acids from C. acaulis.

R2. Q2.

Line 29: full name of the plant

Taxon added.

Line 53: Incorrect citation

References corrected.

Lines 135-139: has already been written in the methodology

Removed as recommended by the Editor.

Line 141: Table 2 not 1

Corrected.

Line 157-158: Latin names of microorganisms - italics

Corrected.

Line 228: „p” italics

Corrected.

A discussion based on 4 references is very modest

In accordance with the Reviewer's recommendations, we expanded the discussion to include additional literature sources.

Line 301: full name

Full names are provided.

Line 316: methanol which concentration

Added in section 4.1. We apologize for the omission. Methanol, like acetonitrile used in this study, was of gradient purity.

Line 316: “ultrasonic bath” detailed equipment data

An ultrasonic bath model and ultrasonic frequency have been added.

Line 317: „evaporated” detailed equipment data

Evaporator model, pressure and evaporation temperature have been added.

Line 318: there are many kinds of filters

The precipitate was separated using filter paper. A relevant annotation was added in the text.

Line 321: „lyophilization” detailed equipment data

Freeze-dryer model, pressure and evaporation time have been added.

Fig 5. hardly legible

The size of the drawing was increased to improve legibility.

Line 366: currently "mol" not "M"

Corrected.

From Line 367: mL

Corrected

Response to R2. Q2. Dear Reviewer, thank you for your detailed review which contributed very significantly to the quality of the manuscript.

Reviewer 3 Report

Comments and Suggestions for Authors

Dear Authors,

This study described the antibacterial activity of fractionated methanolic rude extracts from Carlina acaulis. 

The lack of efforts to address the identification of bioactive compounds from active fractions. I know that this is a significant request; however, I do not see how this manuscript can be published in Molecules if such crucial experiments to identify the bioactive compounds responsible for the observed activity are not included.

Lines 75-93, the lengthy paragraph describing previous studies about antimicrobial assays of crude extracts is hard to digest, a table form of which species, type of post-processed extracts, and biological activities should be included.

Lines 95-99, as suggested that crude extract of C. acaulis has been extensively investigated and demonstrated for the antibacterial activities. Using different polarity solvents or fractionation into different polarity does not change the fact that it was from the crude extract of C. acaulis. Therefore, the justification of this study is invalid (using same crude extract but fractionated into different polarity fractions). 

Table 2, 3 and 4, lacked of positive controls. The figure 2 and 3, lacked of positive controls. It should be included for the publication.

Author Response

Dear Authors,

This study described the antibacterial activity of fractionated methanolic rude extracts from Carlina acaulis.

R3. Q1. The lack of efforts to address the identification of bioactive compounds from active fractions. I know that this is a significant request; however, I do not see how this manuscript can be published in Molecules if such crucial experiments to identify the bioactive compounds responsible for the observed activity are not included.

Response to R3. Q1. Thank you for your valuable feedback on our manuscript. We acknowledge the importance of identifying the specific bioactive compounds responsible for the observed antibacterial activity in fractions B and C. While we haven't included such identification in the current study, we recognize its significance for further development.

This study was designed as an initial investigation of the antimicrobial potential of C. acaulis root extract. Identifying bioactive compounds would require significantly more plant material that is not currently available. C. acaulis is a biennial plant that generates low root mass in comparison with the growth area, thus requiring considerable time and acreage to generate sufficient biomass for detailed analyses. Techniques like bioassay-guided fractionation, mass spectrometry, and NMR can be employed in a follow-up study intended to isolate and identify the active compounds.

The presence of bioactive chlorogenic acids established so far provides a strong initial lead for future study focused on compound identification and purification.

Moreover, chlorogenic acids appear to be the main metabolites responsible for antimicrobial activity, as we pointed out in the discussion.

R3. Q2. Lines 75-93, the lengthy paragraph describing previous studies about antimicrobial assays of crude extracts is hard to digest, a table form of which species, type of post-processed extracts, and biological activities should be included.

Response to R3. Q2. The following table was incorporated into the text, as suggested by the Reviewer.

Species

Extract Type

Activity

Reference

Escherichia coli ATCC 8739

Essential oil

MIC 0.39 µL/mL

[28]

E. coli ATCC 25922

Essential oil

MIC 0.78 µL/mL

[28]

Staphylococcus aureus ATCC 6538

Essential oil

MIC 0.02 µL/mL

[28]

Klebsiella pneumoniae

ATCC 10031

Essential oil

MIC 0.78 µL/mL

[28]

Proteus vulgaris

ATCC 8427

Essential oil

MIC 0.09 µL/mL

[28]

Pseudomonas aeruginosa ATCC 9027

Essential oil

MIC 0.09 µL/mL

[28]

Candida albicans

ATCC 10231

Essential oil

MIC 0.19 µL/mL

[28]

S. aureus

ATCC 6538

Water decoction

MIC 3.1%

[28]

S. aureus

ATCC 6538

Apple cider vinegar decoction

MIC 0.7%

[28]

S. aureus

ATCC 6538

Wine decoction

MIC 1.5%

[28]

MRSA 1000/93

Hexane extract

MIC 0.5 mg/mL

[29]

MRSA ATCC 10442

Hexane extract

MIC 2.0 mg/mL

[29]

VRE ATCC31299

Hexane extract

MIC 0.002 mg/mL

[29]

C. albicans ATCC 90028

Hexane extract

MIC 0.25 mg/mL

[29]

C. glabrata ATCC MYA 2950

Hexane extract

MIC 0.5 mg/mL

[29]

C. albicans ATCC 10231

Essential oil

0.68 mg/mL

[17]

C. albicans ATCC 10231

COx

0.04 mg/mL

[17]

C. albicans ATCC 10231

COx nanoemulsion

1.9 mg/mL

[17]

SARS-CoV-2

COx

IC50 = 234.2 µg/mL in binding to host cell receptor

[30]

R3. Q3. Lines 95-99, as suggested that crude extract of C. acaulis has been extensively investigated and demonstrated for the antibacterial activities. Using different polarity solvents or fractionation into different polarity does not change the fact that it was from the crude extract of C. acaulis. Therefore, the justification of this study is invalid (using same crude extract but fractionated into different polarity fractions).

Response to R3. Q3. We respect the Reviewer's opinion however, we believe that fractionation of extracts, even with known activity, is a good way to find valuable drugs. Vinca alkaloids and paclitaxel were isolated in the same way - the crude extract which was highly cytotoxic was fractionated and then further fractions were fractionated until the pure compound was obtained. We realize that our research did not lead to the isolation of the pure component, however, our planned studies involving the evaluation of the activity of pure chlorogenic acids and their mixtures (in proportions corresponding to those obtained in fraction C) will clarify whether other substances with such activity are present in the fraction in addition to chlorogenic acids. Further fractionation may be necessary to isolate a pure compound that could become a drug candidate.

R3. Q4. Table 2, 3 and 4, lacked of positive controls. The figure 2 and 3, lacked of positive controls. It should be included for the publication.

Response to R3. Q4 Dear Reviewer, original Tables 2 and 3 (new Table 3 and 4) do not actually include positive controls. Antibiotics and synthetic antimicrobial drugs are almost always several times more active than plant extracts and fractions from these extracts.

For example, the MIC of mupirocin is 0.24 × 10-3 mg/mL for S. aureus ATCC 6538 and S. aureus ATCC 43300; 0.49 × 10-3 mg/mL for S. aureus ATCC 25923 and S. epidermidis ATCC 12228; 0.12 × 10-3 mg/mL for B. cereus ATCC 10876. At the time of conducting the research, pure clotrimazole was not available to us and for our own knowledge we determined the sensitivity of the fungi to fluconazole, e.g. MIC for C. albicans was 0.98 × 10-3 mg/mL. We decided that such a comparison would make sense if we isolated pure compounds.

Original Table 4 (new Table 5) includes the control (mupirocin ointment and clotrimazole ointment). Positive controls are included in the figures: vancomycin on Fig. 2, carlina oxide on Fig. 3, mupirocin and clotrimazole ointment on Fig. 4.

Reviewer 4 Report

Comments and Suggestions for Authors

 No further suggestions for the authors, best of wishes in publishing your paper!

Author Response

R4. Q1 No further suggestions for the authors, best of wishes in publishing your paper!

Response to R4. Q1. Thank you very much for your kind words. We are encouraged by your conclusion that no further corrections are needed. This strengthens our confidence in the quality of our work.

Reviewer 5 Report

Comments and Suggestions for Authors

Did the authors perform stability tests with the pharmaceutical preparation?

Did the authors perform dermal toxicity tests?

Fraction c is ethyl acetate, was there any problem with the solubility of this fraction in the pharmaceutical preparation?

The composition of the commercially available amphiphilic ointment base should be mentioned.

The antimicrobial and toxicity effects of the vehicle (commercially available ointment) should be assessed and reported in all tables and figures related to antimicrobial assays.

The antimicrobial effects of chlorogenic acids should be incorporated in the discussion section. The discussion section should be strengthened. The results should be discussed with other studies using pharmaceutical preparations. Is there a possible mechanism of action described for the antimicrobial activity of chlorogenic acids?

The conclusion section should be strengthened. This study has other results to include in this section.

Comments on the Quality of English Language

No comments

Author Response

R5. Q1. Did the authors perform stability tests with the pharmaceutical preparation?

Response to R5. Q1. We appreciate the reviewer raising the important point of stability testing.  In this study, we did not perform stability testing of the ointment. The main reason for this is the early stage of the development. This study focused on the fractionation and the biological evaluation of the material derived from Carlina acaulis L. roots. Preparation of the ointment and the assessment of its properties were proof-of-concept experiments. However, we acknowledge the importance of stability testing for future development of this formulation. We plan to include stability testing in subsequent studies to determine appropriate storage conditions and shelf life.

R5. Q2. Did the authors perform dermal toxicity tests?

Response to R5. Q2. The reviewer raises a valid question regarding dermal toxicity testing. In this study, we did not perform dermal toxicity tests on the pharmaceutical preparation. Currently we only have the capacity to run in vitro safety tests. Experiments carried out using normal human fibroblasts (BJ cell lines) demonstrated better cytotoxicity profile of the ointment with fraction C in comparison to control ointments with mupirocin and clotrimazole (Fig. 4.).

R5. Q3. Fraction c is ethyl acetate, was there any problem with the solubility of this fraction in the pharmaceutical preparation?

Response to R5. Q3. In our formulation process, we did not encounter significant problems with the solubility of Fraction C (ethyl acetate) in the pharmaceutical preparation. This is likely due to the presence of 8% EtOH and 10% DMSO in the final formulation.

R5. Q4. The composition of the commercially available amphiphilic ointment base should be mentioned.

Response to R5. Q4. The composition of Haskobaza was added in section 4.8.

Liquid paraffin                             3.0 (lipophilic properties)

White vaseline                            32.0 (lipophilic properties)

Glyceryl monostearate                3.0 (w/o emulsifier)

Cetostearyl alcohol                      9.0 (emulsifier w/o)

Tween 40                                    7.0 (o/w emulsifier)

Miglyol 812                                 2.0 (emollient)

Propylene glycol                          5.0 (solvent)

Colloidal anhydrous silica             0.1 (stabilizer)

Sorbic acid                                  0.2 (preservative)

Purified water                              to 100.0.

R5. Q5. The antimicrobial and toxicity effects of the vehicle (commercially available ointment) should be assessed and reported in all tables and figures related to antimicrobial assays.

Response to R5. Q5. We agree with the Reviewer that such an approach would be methodologically correct, but we are unable to conduct such research in such a short time. Moreover, an ointment base approved for use in pharmaceutical preparations should not itself demonstrate significant biological activity. However, there is no doubt that the solvent of fraction C (mainly ethanol and DMSO) can have such an effect and the effect of the preparation we propose is the result of the activity of the fraction, its solvent and probably, to a small extent, emulsifiers and sorbic acid contained in Haskobaza. Investigating the influence of individual ointment ingredients on its activity is undoubtedly one of the most important directions of future research and we would like to thank the Reviewer for drawing attention to this issue.

R5. Q6. The antimicrobial effects of chlorogenic acids should be incorporated in the discussion section. The discussion section should be strengthened. The results should be discussed with other studies using pharmaceutical preparations. Is there a possible mechanism of action described for the antimicrobial activity of chlorogenic acids?

Response to R5. Q6. The discussion was strengthened by points suggested by the Reviewer. We are very grateful for this comment as the manuscript benefited greatly from it.

R5. Q7. The conclusion section should be strengthened. This study has other results to include in this section.

Response to R5. Q7. The conclusion section has been completed in accordance with the Reviewer's recommendations.

Round 2

Reviewer 1 Report

Comments and Suggestions for Authors The authors provided answers to some interesting questions regarding the methodology

Author Response

Thank you very much for your kind words. We are encouraged by your conclusion that no further corrections are needed. This strengthens our confidence in the quality of our work.

Reviewer 3 Report

Comments and Suggestions for Authors

The concerns are somewhat been addressed. No further suggestion

Author Response

(The authors gave the same response as above.)

Reviewer 5 Report

Comments and Suggestions for Authors

The authors need more time to perform the experiments that I am requesting.

The antimicrobial and toxicity effects of the vehicle (commercially available ointment) should be assessed and reported in all tables and figures related to antimicrobial assays.

Comments on the Quality of English Language

no comments

Author Response

Dear Reviewer,

Thank you for all the suggestions that enabled us to improve the manuscript. We have added antimicrobial activity data from the ointment base (Tab. 5 and Discussion). Unfortunately, we were not able to perform the study on fibroblasts in the time we received from the Editor (they are currently frozen and it would have taken more than a month to put them into culture and perform the experiment with triplicate). However, if the Reviewer deems it necessary and the Editor gives us more time, we can conduct such an experiment. However, we hope that the Reviewer will take into account the fact that commercially available ointment bases registered as pharmaceutical raw materials must meet safety standards and must not be toxic.